# Creatine Supplementation for Muscle Growth: A Scoping Review of Randomized Clinical Trials from 2012 to 2021

**DOI:** 10.3390/nu14061255

**Published:** 2022-03-16

**Authors:** Shih-Hao Wu, Kuan-Lin Chen, Chin Hsu, Hang-Cheng Chen, Jian-Yu Chen, Sheng-Yan Yu, Yi-Jie Shiu

**Affiliations:** 1Department of Emergency Medicine, China Medical University Hospital, Taichung 404, Taiwan; ambertwu@gmail.com (S.-H.W.); ezn2002@gmail.com (H.-C.C.); 2College of Medicine, China Medical University, Taichung 404, Taiwan; 3Department of Exercise Health Science, National Taiwan University of Sport, Taichung 404, Taiwan; a0987032297@gmail.com (K.-L.C.); jacky880519@gmail.com (J.-Y.C.); j.yan0331@gmail.com (S.-Y.Y.); shiu880511@gmail.com (Y.-J.S.); 4Department of Physical Medicine and Rehabilitation, Yuanrung Hospital Yuansheng Branch, Changhua County 510, Taiwan

**Keywords:** ergogenic aids, performance enhancement, strength, casting, older population, sarcopenia

## Abstract

Creatine supplementation is the most popular ergonomic aid for athletes in recent years and is used for improving sport performance and muscle growth. However, creatine supplementation is not always effective in all populations. To address these discrepancies, numerous studies have examined the use of creatine supplementation for muscle growth. This scoping review aimed to investigate the effects of creatine supplementation for muscle growth in various populations, in which Arksey and O’Malley’s scoping review framework is used to present the findings. For this study, we performed a systematic search of the PubMed, Embase, and Web of Science databases for theses and articles published between 2012 and 2021. A manual search of the reference lists of the uncovered studies was conducted and an expert panel was consulted. Two reviewers screened the articles for eligibility according to the inclusion criteria. Methodological quality was assessed using the National Heart, Lung and Blood Institute’s (NHLBI’s) quality assessment tool. A total of 16 randomized controlled trials (RCTs) were finally included. All the authors extracted key data and descriptively analyzed the data. Thematic analysis was used to categorize the results into themes. Three major themes related to muscle growth were generated: (i) subjects of creatine supplementation—muscle growth is more effective in healthy young subjects than others; (ii) training of subjects—sufficient training is important in all populations; (iii) future direction and recommendation of creatine supplementation for muscle growth—injury prevention and utilization in medical practice. Overall, creatine is an efficient form of supplementation for muscle growth in the healthy young population with adequate training in a variety of dosage strategies and athletic activities. However, more well-designed, long-term RCTs with larger sample sizes are needed in older and muscular disease-related populations to definitively determine the effects of creatine supplementation on muscle growth in these other populations.

## 1. Introduction

Muscle growth involves increasing muscular size, typically through strength training and protein supplementation [1]. Dietary protein intake supports skeletal muscle remodeling after exercise by stimulating muscle protein synthesis and can optimize resistance training-mediated increases in skeletal muscle size and strength [1]. Muscle growth has been an important issue in the sports field for decades. Numerous researchers have focused on how to improve sports performance for athletics [2]. In recent years, muscle growth has also become a popular issue in clinical medicine. Sarcopenia is increasingly linked to aging and is associated with an increased likelihood of adverse outcomes, including falls, fractures, frailty, and mortality [3]. Therefore, finding ways to increase muscle strength and physical performance to prevent falling down, or even fractures, in the older population is one of the most important issues in geriatrics [4].

Numerous supplements for muscle growth have been discovered or developed since the 1980s, such as whey protein, casein, beta-hydroxy beta-methyl butyrate, omega-3 fatty acids, and creatine [5,6,7,8]. Creatine, an ergogenic compound, is an important intermediate in the metabolism of muscles, brain, and other tissues with high energy demand and fluxes [9]. It is endogenously formed from the amino acids arginine, glycine, and methionine in the kidneys and liver [10]. Exogenously, creatine is primarily consumed via meat and/or as a dietary supplement.

Creatine monohydrate supplementation can increase the phosphocreatine/creatine ratio in skeletal muscle tissue, thereby increasing the capacity for rapid adenosine triphosphate (ATP) resynthesis during repeated high-intensity exercise tasks [11,12,13,14]. The increase in lean mass following creatine supplementation has at least partly been attributed to water retention in muscle tissue [13,15]. Greater osmotic pressure following the increase in creatine content has been suggested to result in muscle cell swelling, which is considered a key stimulus for cell growth [13,14,16].

Creatine has also been a popular and effective ergogenic supplementation among athletes of all levels for decades. Despite over 50 years of research, the field of sports nutrition regarding creatine continues to grow at a rapid rate. Many studies have demonstrated that creatine supplementation, in combination with various kinds of training, is effective at augmenting training workouts and increasing muscular strength and lean body mass [14,17]. Due to the sheer volume of studies on creatine supplementation for muscle growth, some evidence of confusion and conflict exists. Our objective was to systematically evaluate the randomized controlled trials published on creatine supplementation for muscle growth in the last 10 years, as well as to identify concepts relevant to creatine supplementation in various populations. We hope that this review will provide more scientific understanding about the safety and efficacy of creatine supplementation in a variety of populations, in addition to recommendations about future research needs.

## 2. Materials and Methods

A scoping review is an appropriate methodology for reviewing large bodies of literature to generate an overview of a research topic [18]. This study was conducted using the five-stage methodological framework for scoping studies developed by Arksey and O’Malley [19]. Guided by this framework, the stages of this scoping review included: (1) identifying the research question; (2) identifying relevant studies; (3) selecting studies; (4) charting the data; (5) collating, summarizing, and reporting the results. The methodologies used for each of the stages within the framework are outlined below. However, an optional stage involving a consultation exercise was excluded in this review.

### 2.1. Identifying the Research Questions

The main purpose of this review was to explore what is known about the effect of creatine supplementation on muscle growth in order to understand the efficacy of creatine supplementation in a variety of populations, and to provide recommendations for future research. Therefore, the research question was purposefully refined to encompass the extensive range and nature of existing research activities in the literature as follows: what is the current available evidence related to creatine supplementation in muscle growth? The following research questions guided this review: (1) What are the dosage strategies of creatine intervention? (2) What are the characteristics of subjects? (3) Are there any training programs in the trials, and what are the programs? (4) What are the outcome measurements and the results? All of the above questions prompted the final key question: can creatine supplementation really help muscle growth?

### 2.2. Identifying Relevant Studies

Relevant published, randomized, controlled trials involving the effect of creatine and muscle were identified in January 2022, using the following online databases: PubMed, Web of Science, and Embase. The following keywords were searched for in the titles: “creatine” AND “muscle.” In addition, “randomized controlled trial” and “recent 10 years” filters were applied to narrow our search. There was no restriction on the original language of the study, and an automatic page translation to English was applied. All of the retrieved studies were manually screened by the authors Wu and Chen for relevance. The studies were saved in reference manager software (EndNote 20, Thomson Reuters, NY, USA).

### 2.3. Selecting Studies

Relevant studies published between 2012 and 2021 were identified during the search. After the removal of duplicates, articles were screened by blinding the results to only show the title and abstract, and then screened using the Endnote referencing software. Articles were included if they (i) mentioned oral creatine supplementation; (ii) were related to muscle strength and muscle mass; (iii) were randomized, placebo-controlled trials. Articles were excluded if there was (i) no exercise or a physical exertion test in the study protocol, (ii) no pure oral creatine supplementation group, (iii) no placebo control, or if they (iv) were gray literature, such as seminar posters or only abstracts or (v) were on animals.

For the articles that met the eligibility criteria, the full article was retrieved and assessed by two independent reviewers to ensure consistent application of the eligibility criteria for inclusion in the review. Disagreements about the study eligibility of the sampled articles were discussed between the two reviewers until a consensus was reached. Because of the present lack of guidelines on the reporting of scoping reviews, the Preferred Reporting Items for Systematic Reviews and Meta-Analysis (PRISMA) guidelines [20] were used to report the flow of the included articles in this review.

### 2.4. Charting the Data

Key items of the information from the included articles were charted onto a form that was developed based on the research question. The key information extracted included the first author’s name, year of publication, study title, study design, number of participants, participant characteristics, creatine dosage, supplementation strategies, supplementation length, exercise modality, exercise program length, methods utilized, outcome measures, and main findings. Two independent reviewers extracted information from each article across each data extraction category. The reviewers met to compare the extracted information; discrepancies were discussed between the two reviewers until a consensus was reached.

#### Assessment of the Quality of the Methodologies of the Studies

Quality Assessment of Controlled Intervention Studies (QACIS) were used to evaluate the methodological quality of the studies included in this scoping review. These assessment tools were developed by the National Heart, Lung and Blood Institute (NHLBI) to assess the risk of bias of the research studies [21]. The studies were assessed using 14 questions cited in the NHLBI tool. For the methodological quality assessment, the items were rated as 1 (meets the criteria), 0 (does not meet the criteria), or N/A (not applicable). The final score for the included articles was calculated based on the total sum of the scored items divided by the total number of items scored, which was expressed as a percentage. Two of the authors (Wu and Chen) independently assessed the quality of the studies, with any disagreements discussed until a consensus was reached (as Appendix A); before this, the Cohen’s kappa score was 0.69, indicating a substantial level of interrater reliability. The NHLBI tool does not have any set threshold for quality scores. Nonetheless, using general study quality assessment guidelines, the studies were characterized as having either low (≤50%), good (51–75%), or excellent (>75%) methodological quality [22].

### 2.5. Collating, Summarizing, and Reporting the Results

After the extraction, separation, grouping, and abstraction of the text findings, two reviewers independently categorized the findings into different groups. Any discrepancies between the two authors over the thematic analysis were clarified by consulting the third author until a consensus on the final results was reached. Additionally, the authors of this review represented different backgrounds, which allows varied perspectives for summarizing the available evidence on the application of creatine supplementation for muscle growth.

## 3. Results

### 3.1. Main Results

Overall, 4086 publications were identified (Figure 1). Of these, 4049 were excluded, leaving 37 publications for potential inclusion (randomized controlled trials and within the last 10 years). Twenty-one of the studies were excluded as either the title mentioned creatine kinase and not creatine supplementation (*n* = 8), there was no placebo control (*n* = 5), pure creatine was not used (*n* = 4), no muscle growth was reported (*n* = 3), or the study was on animals (*n* = 1). In addition, 16 papers were identified as being suitable for inclusion as a result of describing randomized and placebo-controlled trials. The main data and findings from these studies are summarized in Table 1, Table 2, Table 3, Table 4 and Table 5.

### 3.2. Included Studies

Sixteen studies were finally included. All of the trials were conducted between 2012 and 2021. There were a total of 456 participants, with 393 males and 63 females. Twelve trials included only male subjects, with the remaining four trials including both genders. The studies were categorized into five groups according to the different subjects.

### 3.3. Study Outcomes

#### 3.3.1. Healthy Untrained Young Subjects

Healthy untrained young subjects were recruited in 2 of the 16 studies. A study by del Favero et al. used a large dose of creatine supplementation (20 g per day) over a short duration of 10 days. They found that muscle power output and muscle strength increased after squat exercises and bench presses in the creatine group when compared to the placebo control group, even without training [23]. Kaviani et al. implemented a maintenance dose of creatine supplementation (0.07 g/kg per day for 56 days) with whole-body resistance training. This revealed an increase in muscle strength after bench presses, leg presses, and shoulder presses within two weeks in the creatine group when compared to the placebo control group (*p* ≤ 0.05). The muscle strength following tricep extensions had increased by six weeks. Creatine kinase obviously increased in the creatine group when compared to the placebo control group (*p* ≤ 0.05) [24]. In both of these trials, creatine had a positive effect (i.e., an increase) on muscle strength when compared to the placebo control group (Table 1).

#### 3.3.2. Healthy Trained Young Subjects

Healthy trained young subjects were recruited in 6 of the 16 studies. Claudino et al. used a loading dose of creatine supplementation (20 g per day for seven days) and a maintenance dose (5 g per day for 42 days) with whole-body resistance training. This revealed better countermovement jump test performance (*p* = 0.23) and endurance (*p* = 0.05) in the creatine group when compared to the placebo control group [25]. Nunes et al. conducted a loading dose (0.3 g/kg per day for seven days) and a successive maintenance dose (0.03 g/kg per day for 49 days) coupled with whole-body resistance training. This revealed that lean soft tissue increased in the upper limbs, lower limbs, and trunk in the creatine group when compared to the placebo control group (*p* < 0.001), with a greater difference in the upper limbs than in the lower limbs and trunk [26]. Yáñez-Silva et al. implemented a maintenance dose (0.03 g/kg per day for 14 days) with regular elite soccer training. This revealed that the peak power output, mean power output, and total work increased in the creatine group when compared to the placebo control group (*p* ≤ 0.05) [27]. Wang et al. used a large dose of creatine supplementation (20 g per day) and complex training for a short duration of six days. It was found that the muscle strength following bench rows increased in the creatine group when compared to the placebo control group (*p* ≤ 0.05) [28]. Wang et al. used a loading dose (20 g per day for six days) and a maintenance dose (2 g per day successively for 28 days) coupled with complex training. This showed that the muscle strength following half-squats increased in the creatine group when compared to the placebo control group (*p* ≤ 0.05), but there was no significant difference in the countermovement jump test performance. Creatine kinase was clearly decreased in the creatine group when compared to the placebo control group (*p* ≤ 0.05) [29]. Ribeiro et al. used a loading dose (20 g per day for five days) and a maintenance dose (3 g per day successively for 51 days) coupled with whole-body resistance training. It was found that skeletal muscle mass increased in the creatine group when compared to the placebo control group (*p* ≤ 0.05) [15]. In these six studies, creatine had positive effect on muscle strength (i.e., an increase), sport performance, and muscle hypertrophy when compared to the placebo control group (Table 2).

#### 3.3.3. Mimic Immobilization Subjects

Mimic immobilization of the subjects was included in the design of 2 out of the 16 studies. Fransen et al. used a large dose of creatine supplementation (20 g per day) for a short duration of seven days. All subjects received short arm casting for seven days without training. No difference after casting was observed in all work and power data in the ergometric incremental protocol between the creatine and placebo control groups (*p* = 0.57) [30]. Backx et al. used a loading dose (20 g per day for five days) and a maintenance dose (5 g per day successively for 16 days) of creatine supplementation. All of the subjects received long leg casting for seven days without training. This revealed an obvious decrease in leg muscle strength and cross-section area (CSA) of the quadricep muscles after casting in both groups. No differences were noted between groups (*p* = 0.20 and *p* = 0.76, respectively). In the non-immobilized leg, the quadricep muscle CSA did not show significantly different changes between the placebo and creatine group (*p* = 0.63), but 1-RM in the non-immobilized leg increased in the creatine group when compared to the placebo control group during the immobilization period (*p* = 0.03). However, 1-RM in the non-immobilized leg did not change during the subsequent recovery week (*p* = 0.57) [31]. In both of these trials, creatine supplementation had no effect on the preservation of muscle mass or strength during casting (Table 3).

#### 3.3.4. Healthy Untrained Older Subjects

Healthy untrained older subjects were recruited in 4 of the 16 studies. Baker et al. used a large dose of creatine supplementation (20 g for one day) without training. This revealed a decrease in the number of repetitions performed across sets of chest and leg presses, with no differences between the creatine and placebo control groups. Creatine had no effect on the rating of perceived exertion (*p* > 0.05) [32]. Chami et al. divided the subjects into three groups of placebo (PL), high maintenance dose (0.3 g/kg per day, CR-H), and medium maintenance dose (0.1 g/kg per day, CR-M) for 10 days. All subjects received whole-body resistance training. Increased muscle strength and endurance was observed in all groups, but no difference was indicated between the groups (*p* > 0.05) [33]. Candow et al. used a maintenance dose of 0.1 g/kg per day and whole-body resistance training for one year. Muscle thickness was measured using dual-energy X-ray absorptiometry (DXA). After 12 months of training, both groups experienced similar changes in muscle thickness and strength following chest presses and hack squats. No difference was noted between the creatine and placebo groups [34]. Candow et al. also adopted a maintenance dose (0.1 g/kg per day) with whole-body resistance training for one year. This time, muscle and bone density were measured using peripheral quantitative computed tomography (pQCT). After 12 months of training, the lower leg muscle density increased in the creatine group (Δ + 0.83 ± 1.15 mg·cm^−3^; *p* = 0.016) when compared to the placebo group (Δ –0.16 ± 1.56 mg·cm^−3^), but no changes were noted in the forearm muscle [35]. In these four trials, creatine supplementation had no obvious effect on muscle strength, sport performance, or muscle hypertrophy in healthy untrained older subjects. However, creatine supplement may have some favorable effects on the muscle density of the lower limbs (Table 4).

#### 3.3.5. Subjects with Diseases

Subjects with diseases were included in 2 of the 16 studies. Domingues et al. used a loading dose of creatine supplementation (20 g per day for seven days) and a successive maintenance dose (5 g per day for 49 days) without training in patients with symptomatic peripheral arterial disease. There were no significant differences found in the function capacity by total walking distance, whether post-loading or post-maintenance (*p* = 0.170) [36]. In their study of patients with juvenile dermatomyositis, Dover et al. adopted a maintenance dose (for patients < 40 kg: 0.15 g/kg per day; for patients > 40 kg: 4.69 g/m^2^ per day, for 28–180 days) without training. No significant changes were found in muscle function, strength, aerobic capacity, disease activity, fatigue, physical activity, or quality of life between the creatine and placebo groups, and no significant adverse effects were observed [37]. In both of these trials, there was no obvious observed effect on muscular performance in the creatine group when compared to the placebo group (Table 5).

### 3.4. Three Major Themes That Affect Muscle Growth Due to Creatine Supplementation

#### 3.4.1. Theme 1: Subjects of Creatine Supplementation

Creatine had a positive effect on muscle strength (i.e., an increase), sport performance, and muscle hypertrophy in all healthy young populations, even in those who were untrained [15,23,24,25,26,27,28,29]. However, only Candow et al. found that lower leg muscle density also increased in a creatine group of older subjects [35]. None of the other trials in this review observed a positive effect of creatine on muscle growth in other populations. Therefore, creatine supplementation appears to be more effective for muscle growth in healthy young subjects than other populations.

#### 3.4.2. Theme 2: Training of Subjects

All of the above healthy young populations, even the untrained, received sufficient training just before or during creatine supplementation [15,23,24,25,26,27,28,29]. None of the other trials revealed a positive effect of creatine on muscle growth, except that of Candow et al., in which lower leg muscle density increased in the creatine group [35]. This might be due to the lower limb training volume (including daily activity, such as walking) being greater than in the upper limbs. Therefore, sufficient training is important in all populations.

#### 3.4.3. Theme 3: Future Direction and Recommendations for Research on Creatine Supplementation for Muscle Growth

There is still a limited number of research studies in casting and other muscular-related diseases. Moreover, the results regarding the effect of creatine supplementation on muscle damage or recovery in the work of Wang et al. and Kaviani et al. [24,29] are conflicting. Future research regarding creatine supplementation for muscle growth could focus on injury prevention and utilization in medical practice.

## 4. Discussion

The aim of this scoping review was to determine the effect of supplementary creatine on improving muscle growth in various populations. Creatine supplementation showed clear benefits in young training adults but ambiguous effects in other untrained groups. The dosage, type of training, age of subjects, and presence of diseases are variables that affect muscle growth under creatine supplementation.

### 4.1. Dosage Strategies of Creatine Supplementation

The positive effects of creatine supplementation on muscle strength, muscle mass, and sport performance seemed to only be found in healthy young adults, regardless of whether they were trained or untrained before use, and different dosage strategies were effective. In the studies of young adults, creatine supplementation without a loading dose still had a positive effect on muscle mass, sport performance, and muscle strength within two weeks [24]. Consistent with this, Antonio et al. showed increased intramuscular creatine storage, muscle accretion, and muscle performance even under lower daily dosages of creatine supplementation (i.e., 3–5 g/day) [17].

### 4.2. Exercise and Its Relationship with Creatine Supplementation

#### 4.2.1. Type of Exercise That Benefits from Creatine Supplementation

Among the trials on healthy trained young subjects, resistance training, elite soccer training, canoe basic training, and plyometric training were included [15,25,26,27,28,29]. The above information revealed that all kinds of assessed exercise and training with creatine supplementation were effective for increasing muscle accretion and muscle performance. Consistent with this, Antonio et al. mentioned that a variety of athletic activities, not just those involving resistance/power, may benefit from creatine supplementation [17].

#### 4.2.2. Without Exercise during the Creatine Supplementation Period

The healthy untrained subjects in the trial of del Favero et al. were instructed to refrain from any exercise training program throughout the study. However, those subjects with creatine supplementation still had increased muscle power output and muscle strength when compared to the placebo. According to the study protocol, a 12 day exercise program, including a familiarization session, 1-RM maximum dynamic strength test, and muscle power output test, was executed prior to creatine supplementation [23]. It seems that the 12 day exercise program was enough to observe muscles benefiting from creatine supplementation, including through an increase in intramuscular creatine storage and muscle growth as well as a further increase in muscle power output and muscle strength. In the trial of Backx et al., the non-immobilized leg (without training, but needing daily physical activity) and the quadricep muscle CSA did not change differently in the placebo or creatine group (*p* = 0.63) during immobilization. However, 1-RM of the non-immobilized leg increased during the immobilization period (*p* = 0.03) with no differences between the placebo and creatine groups (*p* = 0.90) [31]. Therefore, creatine supplementation was ineffective on muscle growth when sufficient training was lacking.

### 4.3. The Effect of Creatine Supplementation on Muscle Damage or Recovery

Serum creatine kinase activity reflects muscle membrane disruption, which is often elevated after exercise or resistance training. Wang et al. found that creatine kinase was obviously decreased in the creatine group when compared to the placebo control group (*p* ≤ 0.05). Creatine supplementation could reduce muscle damage after training [29]. However, Kaviani et al. found that creatine kinase more obviously increased in the creatine group when compared to the placebo control group (*p* < 0.05) after whole-body resistance training. Creatine supplementation did not prevent muscle damage [24]. Thus, the results are conflicting. Jiaming et al. performed a systematic review and meta-analysis of RCTs for the effect of creatine supplementation on recovery following exercise-induced muscle damage. This revealed that creatine supplementation is effective in reducing the immediate muscle damage that happens < 24, 24, 48, 72, and 96 h post-exercise. In the current meta-analysis, the positive effects of creatine could cause a decrease in the overall CK concentration. However, due to high heterogeneity and the medium risk of bias in articles, they suggest that these results are taken into account and that the data be interpreted with caution [38]. Northeast et al. also performed a systematic review and meta-analysis of human intervention trials regarding the effect of creatine supplementation on the markers of exercise-induced muscle damage. They found that creatine supplementation does not accelerate recovery following exercise-induced muscle damage. Creatine attenuated creatine kinase activity only at 48 h post-exercise and not at any other time points. High (I^2^; >75%) and significant (chi^2^; *p* < 0.01) heterogeneity was identified for all outcome measures at various follow-up times [39]. Therefore, the effect of creatine supplementation on muscle damage is still controversial.

### 4.4. The Effect of Creatine Supplementation on Mitigating or Attenuating the Loss of Muscle Mass or Muscular Strength during Immobilization

Creatine supplementation while immobilized during casting is a research theme that has been emerging in recent years. Fransen et al. and Backx et al. found that creatine supplementation had no effects on the preservation of muscle mass or strength during casting [30,31]. However, Harmon et al. found that creatine supplementation may promote maintenance and mitigate the loss of muscle mass, muscular strength, and endurance, as well as promote healthy glucoregulation, during periods of immobilization [40]. However, different study protocols (e.g., involved muscle group, duration, and creatine dose) may have influence the observed findings [40]. Therefore, the effect of creatine supplementation on mitigating or attenuating the loss of muscle mass or muscular strength during immobilization is still controversial.

### 4.5. The Effect of Creatine Supplementation on Older Populations

In healthy untrained older subjects, creatine supplementation had no obvious effect on the increase in muscle strength, sport performance, or muscle hypertrophy when compared to the placebo control group. However, Candow et al. published two review articles in 2014 and 2019, mentioning that creatine may positively increase muscle mass and strength in the elderly [41,42]. However, Candow et al. found that after 12 months of training, there was no obvious increase in muscle thickness by DXA in the creatine group when compared to the placebo group [34]. Candow et al. performed another trial using pQCT for muscle density and showed that creatine supplementation may have some favorable effects on muscle density in the lower but not upper limbs [35]. In the recent two studies of Candow et al., which measured muscle mass differently, the authors were attempting to test a different method to find evidence that creatine promotes muscle growth. A review article by Antonio et al. mentioned that creatine supplementation in the elderly population may have a positive effect on increasing muscle mass, especially during resistance training [17]. Based on the current evidence, it is not clear whether creatine can increase muscle mass and strength in the elderly population, and more research is needed to reach a clear conclusion.

### 4.6. The Effect of Creatine Supplementation on Patients

Many diseases may cause muscle atrophy or muscle lesions and result in a loss of muscle strength and a decrease in quality of life. Creatine supplementation is a novel field in clinical medicine. Domingues et al. and Dover et al. revealed no obvious effect on muscular performance in the creatine group when compared to the placebo group in peripheral arterial disease and juvenile dermatomyositis, respectively. Moreover, the subjects had high compliance, with almost no side effects due to creatine supplementation. Thus, creatine appears to be safe for the human body and for all populations, with no additional adverse effects on liver and kidney function [17,43,44].

### 4.7. Strengths and Limitations

In this scoping review, a rigorous, comprehensive search strategy across three databases was used to identify relevant articles conforming to the study criteria. Furthermore, a systematic, in-depth data extraction process was performed in duplicate to ensure reliability. Despite these strengths, this review has several limitations. First, the main limitation of this literature review is in the selection bias. There are many related literature works that do not have the word “muscle” in the title and may include “resistance training” or “performance.” However, in using the “creatine” and “muscle” search keywords and a ten-year period, this study should have uncovered sufficient studies to represent the current trend in the research on creatine supplementation for muscle growth. Second, due to creatine supplementation for muscle growth being a scientific issue, intervention effectiveness and trial quality are more important than the opinion of experts. Therefore, we did not include other types of papers, which reduced the breadth and depth of the articles included in this scoping review. Additionally, unlike that suggested by Arksey and O’Malley [19], we did not consult with stakeholders, which could have resulted in an overemphasis of the gaps that need to be filled. However, approximately 60% of all scoping reviews do not include consultations with stakeholders [45]. Lastly, the results of this study may have been influenced by the search terms that were used, the number of databases searched, and the selection of databases used in the search. As a result, the findings of this review may be influenced by publication bias.

## 5. Conclusions

This scoping review provides a current map of the literature on creatine supplementation in muscle growth in various populations. Creatine is an efficient supplementation for increasing muscle strength, muscle mass, and athletic performance in the healthy young population with adequate training in a variety of dosage strategies and athletic activities. However, high-level evidence-based research on the efficacy and validity of creatine supplementation in muscle growth for the elderly or patients with muscle-related diseases is lacking. In addition to the utilization of creatine supplementation in the older population with sarcopenia and in patients with muscle-related diseases, this scoping review identified creatine supplementation with resistance training for muscle wasting in patients with cancer, end stage renal disease, and heart failure as areas for potential therapeutic research and future exploration.

## Figures and Tables

**Figure 1 nutrients-14-01255-f001:**
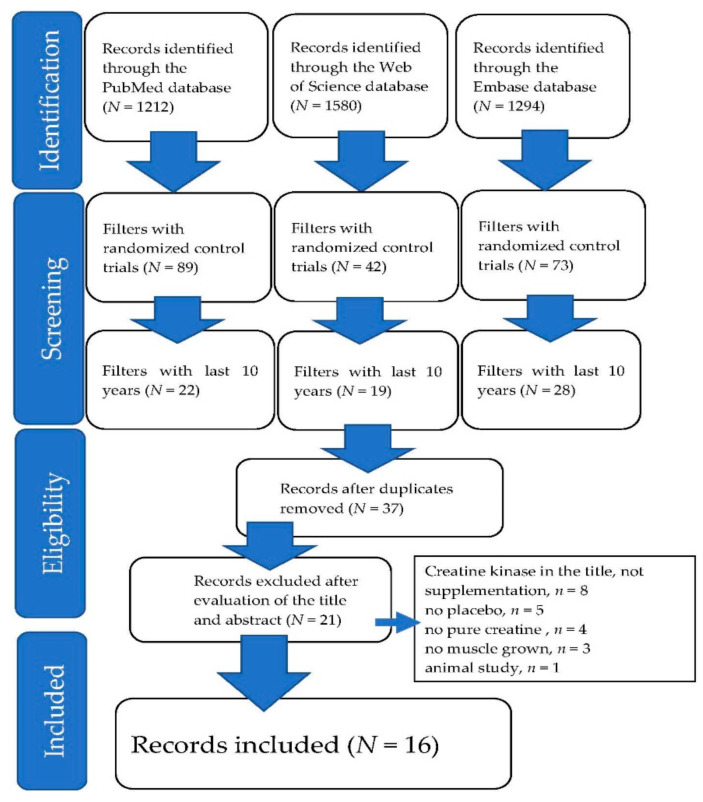
Preferred Reporting Items for Systematic Reviews and Meta-A (PRISMA) flow diagram.

**Table 1 nutrients-14-01255-t001:** Studies recruiting healthy untrained young subjects.

Authors (Year)	Design	QACIS Score	Participants	Creatine Dose (g/day)	Duration (Days)	Training Exercise	Evaluation Exercise	Outcome Measures (Muscle-Related)	Main Findings (Muscle-Related)
del Favero et al. [23]	RP DB	85.71	34 healthy untrained males 18–30 years old	2 × 10 g	10	N/A	Squat exercise Bench press	Muscle power output by a linear encoder 1RM	Muscle power output: Squat exercise: CR > PL (*p* = 0.003) Bench-press: CR > PL (*p* = 0.039) 1RM in CR: Post > pre Squat exercise: *p* = 0.027 Bench-press: *p* < 0.0001 No change in PL
Kaviani et al. [24]	RP DB	85.71	18 healthy untrained males 23 ± 3 years old	0.07 g/kg	56	Bench press Leg press Biceps curl Tricep extension Shoulder press Lat pull-down	1RM of left exercise	Check 1RM every two weeks for eight weeks CK	1RM CR > PL within two weeks after bench, leg, and shoulder presses (*p* < 0.05) By six weeks, 1RM CR > PL in above three items and tricep extension (*p* < 0.05) Eight weeks, then same as six weeks, no significant difference in biceps curls and lat pull-downs CK, CR > PL (*p* ≤ 0.05)

RP: randomized parallel; DB: double blind; N/A: not applicable; RM: repetition maximum; CR: creatine; PL: placebo; CK: creatine kinase.

**Table 2 nutrients-14-01255-t002:** Studies recruiting healthy trained young subjects.

Authors (Year)	Design	QACIS Score	Participants	Creatine Dose (g/day)	Duration (Days)	Training Exercise	Evaluation Exercise	Outcome Measures (Muscle-Related)	Main Findings (Muscle-Related)
Claudinoet al. [25]	RP DB	71.43	23 healthy male elite soccer player 18.3 ± 0.9 years old	4 × 5 g × 7 days 5 g × 42 days	49	Whole-body resistance training Jumping Specific training for soccer	Countermovement jump (CMJ) test	Countermovement jump test	CMJ test: CR > PL (*p* = 0.23) Reduction in jump performance, PL > CR (*p* = 0.05)
Nunes et al. [26]	RP DB	85.71	43 healthy trained males 22.7 ± 3 years old	0.3 g/kg × 7 days 0.03 g/kg × 49 days	56	Whole-body resistance training	N/A	DXA for lean soft tissue	↑ Lean soft tissue (LST) in upper limbs, lower limbs, and trunk, CR > PL (*p* < 0.001) ULLST (7.1 + 2.9%) > LLLST (3.2 + 2.1%) and TLST (2.1 + 2.2%).
Yáñez-Silva et al. [27]	RP DB	71.43	20 healthy trained male elite soccer players 17.0 ± 0.5 years old	0.03 g/kg	14	Soccer training for skill development and anaerobic and aerobic performance development	Wingate anaerobic test	Peak power output (PPO), mean power output (MPO), fatigue index (FI), and total work	↑ PPO, MPO, and total work in CR vs. PL (*p* ≤ 0.05)
Wang et al. [28]	RP DB	85.71	17 healthy trained male high school canoeists 16.6 years old	4 × 5 g × 6 days	6	Bench row Complex training bouts Overhead medicine ball throw test	As left	1RM of bench row Complex training bouts to determine the optimal individual timing of PAP Distance of overhead medicine ball throw test	↑ Maximal strength, CR vs. PL (*p* ≤ 0.05) ↓ Optimal individual PAP time, CR vs. PL (*p* ≤ 0.05) No effect on explosive power
Wang et al. [29]	RP DB	92.86	30 healthy trained males 20 ± 2 years old	4 × 5 g × 6 days 2 g × 28 days	34	Complex training: Half-squat Vertical jump Squat jump	Body composition 1RM half-squat 30 m sprint and CMJ test	CMJ test Sprint test 1RMCK	1RM, CR > PL (*p* ≤ 0.05) CK, CR < PL (*p* ≤ 0.05) CMJ test: No significant difference in CR or PL
Ribeiro et al. [15]	RP DB	85.71	30 healthy trained male elite soccer players 21.8 ± 4.2 years old	4 × 5 g × 5 days 3 g × 51 days	56	Whole-body resistance training	N/A	Bioelectrical impedance device for ICW and ECW DXA for skeleton muscle mass	↑ Skeletal muscle mass, total body water, intracellular water in CR vs. PL (*p* ≤ 0.05)

RP: randomized parallel; DB: double blind; N/A: not applicable; RM: repetition maximum; CR: creatine; PL: placebo; ICW: intra-cellular water; ECW: extra-cellular water; CK: creatine kinase; DXA: dual-energy X-ray absorptiometry; ↑: increase; ↓: decrease.

**Table 3 nutrients-14-01255-t003:** Studies recruiting healthy mimic immobilization subjects.

Authors (Year)	Design	QACIS Score	Participants	Creatine Dose (g/day)	Duration (Days)	Training Exercise	Evaluation Exercise	Outcome Measures (Muscle-Related)	Main Findings (Muscle-Related)
Fransen et al. [30]	RP DB	92.86	25 healthy untrained males/females 24 ± 4 years old Short arm casting for seven days	20 g	7	N/A	Forearm wrist flexion ergometer	Ergometry incremental protocol to fatigue and two constant load (CL1 and CL2) exercise bouts Magnetic resonance spectroscopy (MRS) for PCr	No difference in any work and power data, after casting, CR vs. PL (*p* = 0.57) Work production in CL1 tended (*p* = 0.073) to attenuate in CR vs. PL, no difference in CL2 ↓ PCr after casting in PL (*p* = 0.003), no change in C R (*p* = 0.31)
Backx et al. [31]	RP DB	92.86	27 healthy untrained males 23 ± 1 years old Long leg casting for seven days	20 g × 5 days 5 g × 16 days	21	N/A	Knee extension	Muscle biopsy Computed tomography (CT) for cross-sectional area (CSA) 1RM	↓ Quadricep muscle CSA after casting, no differences between groups (*p* = 0.76) ↓ Leg muscle strength, after casting, no differences between groups (*p* = 0.20) When non-responders to creatine loading were excluded (*n* = 6), responders (*n* = 8) still showed no signs of preservation of muscle mass or strength during casting

RP: randomized parallel; DB: double blind; N/A: not applicable; RM: repetition maximum; CR: creatine; PL: placebo; ↑: increase; ↓: decrease.

**Table 4 nutrients-14-01255-t004:** Studies recruiting healthy untrained old subjects (to be continued).

Authors et al.(Year)	Design	QACIS Score	Participants	Creatine Dose (g/day)	Duration (Days)	Training Exercise	Evaluation Exercise	Outcome Measures (Muscle-Related)	Main Findings (Muscle-Related)
Baker et al. [32]	RC DB	85.71	Nine healthy untrained males 54.8 ± 4.3 years old	20 g	1	N/A	Leg press Chest press Cycle ergometer	1RM Muscle endurance test (three sets at 70% baseline 1RM to muscle fatigue; 1 min rest between sets)	↓ In the number of repetitions performed across sets, with no differences between CR and PL (*p* > 0.05) CR had no effect on the rating of perceived exertion (leg press: CR, 9.1 ± 0.7; PL, 8.9 ± 0.6; chest press: CR, 9.0 ± 1.0; PL, 8.8 ± 0.8) (*p* > 0.05)
Chami et al. [33]	RP DB	92.86	33 healthy untrained males 58.5 ± 4.7 years old	0.3 g/kg 0.1 g/kg	10	N/A	Leg press Chest press Handgrip Walk backward as fast as possible on an elevated board	1RM Muscle endurance (maximal number of repetitions performed for one set at 80% and 70% 1RM for the leg and chest presses, respectively) Physical performance (balance and falls)	↑ Muscle strength and endurance in all groups, but no difference in groups CR-H, CR-M, and PL (*p* > 0.05)
Candow et al. [34]	RP DB	92.86	35 healthy untrained males 56.9 years old	2 × 0.05 g/kg	365	Resistance training three times per week for one year	Chest press Hack squat	Maximal strength DXA	After 12 months of training, both groups experienced similar changes in muscle thickness and muscle strength, with no difference between CR and PL
Candow et al. [35]	RP DB	78.57	70 healthy untrained males/females 58 ± 6 years old	0.1 g/kg	365	Resistance training three times per week for one year	N/A	pQCT	↑ Lower leg muscle density in CR (Δ + 0.83 ± 1.15 mg·cm^−3^; *p* = 0.016) compared to PL (Δ − 0.16 ± 1.56 mg·cm^−3^),with no changes in the forearm muscle

RP: randomized parallel; RC: randomized crossover; DB: double blind; N/A: not applicable; RM: repetition maximum; CR: creatine; PL: placebo; DXA: dual-energy X-ray absorptiometry; pQCT: peripheral quantitative computed tomography; ↑: increase; ↓: decrease.

**Table 5 nutrients-14-01255-t005:** Studies recruiting disease-related subjects.

Authors (Year)	Design	QACIS Score	Participants	Creatine Dose (g/day)	Duration (Days)	Training Exercise	Evaluation Exercise	Outcome Measures (Muscle-Related)	Main Findings (Muscle-Related)
Domingues et al. [36]	RP DB	85.71	29 symptomatic PAD males/females 64 ± 8 years old	4 × 5 g × 7 days 5 g × 49 days	56	N/A	Six min walk test	Functional capacity (total walking distance) was assessed by the six min walk test Calf muscle StO_2_ was assessed through near-infrared spectroscopy	No significant differences were found for function capacity (PL: Pre 389 ± 123 m vs. post-loading 413 ± 131 m vs. post-maintenance 382 ± 99 m; CR: Pre 373 ± 149 m vs. post-loading 390 ± 115 m vs. post-maintenance 369 ± 115 m, *p* = 0.170) and the calf muscle StO_2_ parameters (*p* > 0.05)
Dover et al. [37]	RC DB	85.71	13 juvenile dermatomyositis males/females 13 (7–14) years old	< 40 kg → 0.15 g/kg > 40 kg → 4.69 g/m^2^	28–180	N/A	Wingate anaerobic test Cycle ergometer submaximal test to measure aerobic capacity Maximal jump test Handgrip strength test	Muscle function Aerobic capacity Muscle strength Fatigue Physical activity Quality of life assessed by questionnaires	No significant changes in muscle function, strength, aerobic capacity, disease activity, fatigue, physical activity or quality of life in CR vs. PL No significant adverse effects

RP: randomized parallel; RC: randomized crossover; DB: double blind; PAD: peripheral arterial disease; N/A: not applicable; CR: creatine; PL: placebo.

## Data Availability

No new data were created or analyzed in this review. Data sharing is not applicable for this article.

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
