# Peer review of "Creatine Supplementation for Muscle Growth: A Scoping Review of Randomized Clinical Trials from 2012 to 2021"

_nutrients, 2022, doi:10.3390/nu14061255_

Round 1

Reviewer 1 Report

Dear authors,

I would like to thank you for writing a really comprehensive scoping review on creatine and muscle growth. The manuscript is well-written and organized. I have some minor comments which are listed blow.

Line 39- What do you mean by " increase in muscle fibre? Please elaborate on this, perhaps you can talk more on structural and contractile protein in muscle and how they change in response to creatine and exercise. 

Line 39- "More muscle mass leads....... muscular size." please re-word this as you are stating something obvious. You also need to provide references for this statement. 

It would be worth mentioning something related to dietary intake in the selected papers as it is a significant contributing factor in muscle mass. 

Author Response

1.Line 55 to Line 61 had described on structural and contractile protein in muscle and how they change in response to creatine and exercise

2. Thank you four your opinion. ï¼·e have revised by your suggestion.

3. Thank you four your opinion. ï¼·e have revised by your suggestion.

Reviewer 2 Report

Thank you ever so much for the opportunity to review your work. It is very well written and easy to follow. Really the only thing I can think of is a very critical review of those studies and you conclusion is very positive although you mention limitations. There are very few patients overall, this would suggest the effect of creatinine is very powerful indeed. One remains sceptical as as you say the evidence is of low quality and it would help for you to describe the limitations of the techniques etc used to help less experienced readers and to demonstrate your extensive knowledge?

Author Response

 I am so sorry to let you feel that. I have reviewed this article several times, I think it should be this sentence : "  However, high-level evidence-based research on the efficacy and validity of creatine supplementation in muscle growth for the elderly or patients with muscle-related diseases is lacking."(line 452) What I mean is the number of articles was lacking, not the quality was poor. All of the trials for elder population and disease related population were excellent by QACIS score in this scoping review!  

I have revised the abstract by your suggestion.

Thank you very much !